# Overestimation of Oxygen Saturation Measured by Pulse Oximetry in Hypoxemia. Part 1: Effect of Optical Pathlengths-Ratio Increase

**DOI:** 10.3390/s23031434

**Published:** 2023-01-28

**Authors:** Eyal Elron, Ruben Bromiker, Ori Gleisner, Ohad Yosef-Hai, Ori Goldberg, Itamar Nitzan, Meir Nitzan

**Affiliations:** 1Neonatal Intensive Care Unit, Schneider Children’s Medical Center, Petah Tikva 4920235, Israel; 2Sackler Faculty of Medicine, Tel Aviv University, Tel Aviv 6997801, Israel; 3Lev Academic Center, Jerusalem 9116001, Israel; 4Pediatric Pulmonology Institute, Schneider Children’s Medical Center, Petach Tikva 4920235, Israel; 5Shaare Zedek Medical Center, Jerusalem 9103102, Israel; 6Department of Pediatrics, Hebrew University of Jerusalem Medical School, Jerusalem 9112102, Israel

**Keywords:** arterial blood oxygen saturation, pulse oximetry, hypoxemia, occult hypoxemia, near infrared, inaccuracy, pathlengths-ratio

## Abstract

On average, arterial oxygen saturation measured by pulse oximetry (SpO_2_) is higher in hypoxemia than the true oxygen saturation measured invasively (SaO_2_), thereby increasing the risk of occult hypoxemia. In the current article, measurements of SpO_2_ on 17 cyanotic newborns were performed by means of a Nellcor pulse oximeter (POx), based on light with two wavelengths in the red and infrared regions (660 and 900 nm), and by means of a novel POx, based on two wavelengths in the infrared region (761 and 820 nm). The SpO_2_ readings from the two POxs showed higher values than the invasive SaO_2_ readings, and the disparity increased with decreasing SaO_2_. SpO_2_ measured using the two infrared wavelengths showed better correlation with SaO_2_ than SpO_2_ measured using the red and infrared wavelengths. After appropriate calibration, the standard deviation of the individual SpO_2_−SaO_2_ differences for the two-infrared POx was smaller (3.6%) than that for the red and infrared POx (6.5%, *p* < 0.05). The overestimation of SpO_2_ readings in hypoxemia was explained by the increase in hypoxemia of the optical pathlengths-ratio between the two wavelengths. The two-infrared POx can reduce the overestimation of SpO_2_ measurement in hypoxemia and the consequent risk of occult hypoxemia, owing to its smaller increase in pathlengths-ratio in hypoxemia.

## 1. Introduction

Oxygen supply to the organs is essential for the metabolism of their cells, where oxygen and glucose react with adenosine diphosphate (ADP) to produce adenosine triphosphate (ATP), which is the source of energy for cellular reactions. The level of the oxygen supply to the tissue depends on two factors: the oxygen concentration in the arterial blood and blood perfusion to the tissue. While there is no accepted technique for routine clinical measurement of tissue blood perfusion, arterial-blood oxygen concentration is routinely estimated by arterial-blood oxygen saturation (SaO_2_), the ratio between the concentrations of oxyhemoglobin and total hemoglobin (the combined amount of oxyhemoglobin (HbO_2_) and deoxyhemoglobin (deHb) concentrations). SaO_2_ can be measured invasively in extracted arterial blood samples by CO-oximetry, or non-invasively, by means of pulse oximetry. The invasive technique is considered the gold standard for the measurement of arterial blood oxygen saturation, whereas non-invasive pulse oximetry only provides an estimate of SaO_2_ because in many situations, significant discrepancies can be found between actual SaO_2_, measured by CO-oximetry in extracted blood, and arterial oxygen saturation estimated by pulse oximetry. The latter is referred to as SpO_2_ because of that potential discrepancy. 

The level of accuracy (in fact, inaccuracy) of a pulse oximeter in a clinical study is determined by the extent of the disparity of the individual differences between SpO_2_ and SaO_2_ measurements, estimated by the bias—the *mean* value of the individual SpO_2_−SaO_2_ differences, and by the standard deviation (SD) of those differences. Manufacturers of pulse oximeters generally claim an accuracy of 2%, which means that the SD of the SpO_2_SaO_2_ differences is 2%. A standard deviation of 2% is associated with an expected error of 4% (two SDs) or more among 5% of the examinations [1,2]. The clinical meaning of an expected error of 4% was recently demonstrated in the FDA Communication titled: “Pulse Oximeter Accuracy and Limitations: FDA Safety Communication” [2]. Under certain circumstances, inaccurate measurements by pulse oximeters might lead to occult (hidden) hypoxemia—low levels of oxygen saturation in arterial blood that was not recognized by SpO_2_, because it overestimated SaO_2_. For a single examination on an individual patient, a pulse oximetric reading of 90% only indicates that oxygen saturation in the arterial blood may range between 86% and 94%, with a level of confidence of 95% [2].

The error in SpO_2_ measurement is even greater in hypoxemia and in dark skin: in addition to greater SD, there is also positive bias [3,4,5,6,7]. The overestimation of SpO_2_ measurement in hypoxemia and in dark skin might increase the risk of occult (hidden) hypoxemia, generally defined as SpO_2_ ≥ 92% while SaO_2_ ≤ 88%, which might lead to the deprivation of proper treatment. Despite the large number of studies that describe the overestimation of SpO_2_ readings in hypoxemia and in dark skin, the origins of these effects have not yet been established. In this article, the theory of pulse oximetry will be described in order to explain the overestimation of SpO_2_ readings in hypoxemia by an increase in the optical pathlengths-ratio between the two wavelengths. Understanding the origin of SpO_2_ overestimation in hypoxemia is a key to solving the problem, or at least mitigating it. 

## 2. Background 

### 2.1. Overestimation of SpO_2_ in Hypoxemia

Clinical studies for the measurement of SpO_2_ accuracy/error, assessed by the mean and by the SD of the individual SpO_2_−SaO_2_ differences, have been performed on various groups of patients who needed arterial blood gas readings for medical purposes, and the majority of those studies have yielded a positive bias in hypoxemia that decreased for higher values of SaO_2_ [3,6,8,9,10,11,12,13]. In the SaO_2_ range of 90–100%, some studies demonstrated negative values of the bias [8,11,12,13,14], while other studies showed an overestimation of SpO_2_ [3,15]. Of particular significance are hypoxemic infants with cyanotic congenital heart disease (CHD), whose skin exhibits a bluish tint due to the high level of deoxygenated hemoglobin in their arterial and venous blood. Similarly to adults, the cyanotic CHD infants demonstrated significant positive values for the SpO_2_−SaO_2_ difference, which decreased towards greater values of SaO_2_ [4,5,16,17,18,19,20,21,22,23]. 

The magnitude of the SpO_2_−SaO_2_ difference and the decreasing slope of the regression line of that difference versus SaO_2_ vary among the studies, depending on several variables including the commercial model of the pulse oximeter [5,8,9,21], the disease [6,24], skin color [8,9,12,23], and sensor location [17].

In several studies, SpO_2_ was simultaneously measured using different pulse oximeter models from Masimo and Nellcor and was compared to invasive SaO_2_ readings. In the study by Harris [5], the decreasing slope of the regression line of the SpO_2_−SaO_2_ difference versus SaO_2_ was significantly steeper in the Nellcor device (Nellcor N-600 with Max-I sensor, Nellcor; Medtronic, Dublin, Ireland) than in the two Masimo devices (Masimo Standard and Masimo Blue, Masimo, Irvine, CA, USA). In the studies of Bickler [8] and Feiner [9], Nellcor (Nellcor N-595, Nellcor Inc., Pleasanton, CA, USA) and Nonin (Nonin 9700, Nonin Inc., Plymouth, MN pulse oximeters showed a *decreasing* slope of SpO_2_−SaO_2_ as a function of SaO_2_, while Masimo Radical (Masimo Inc., Irvine, CA, USA) showed an *increasing* slope. In the study by Kim et al. [4], similar decreasing slopes of the SpO_2_−SaO_2_ difference as a function of SaO_2_ were found for two Masimo models and a Nellcor model.

The level of overestimation of SpO_2_ in hypoxemia also depends on the disease that causes the hypoxemia. COVID-19, as a pulmonary disease, is a risk factor for hypoxemia, and in a study by Nguyen et al. [6] on ICU patients with acute respiratory distress syndrome (ARDS), the bias was 0.3% for non-COVID-19 and 2% for COVID-19 patients. It should be noted that in another study on hypoxemic COVID-19 patients, the mean SpO_2_ was found to be *lower* than the mean SaO_2_ by 4.02% [25]; similar results were obtained by Wilson-Baig [26].

In this article, a possible explanation of the overestimation of SpO_2_ readings in hypoxemia will be proposed. To explain our hypothesis that the disparity between SpO_2_ and SaO_2_ in hypoxemia is related to the increase in hypoxemia of pathlengths-ratio between the two wavelengths of the pulse oximeter, a description of the pulse oximetry theory and its relevant assumptions will be presented. 

### 2.2. Pulse Oximetry

Oximetry is an optical technique that utilizes the differing light absorption of oxyhemoglobin and deoxyhemoglobin for the measurement of oxygen saturation in the blood. Figure 1 presents the spectral curves of the extinction coefficient of HbO_2_ and deHb—the specific absorption constant—which is defined by the absorption constant per molar of hemoglobin. Pulse oximetry is a technique that utilizes the photoplethysmographic (PPG) pulses, the pulses of light transmitted through tissue, caused by cardiac-induced oscillations in arterial blood volume (Figure 2). Pulse oximetry is based on the measurement of the PPG signal by two suitable wavelengths and the determination of the SpO_2_ by the analysis of the PPG signals. The use of the PPG signal in pulse oximetry enables the distinction between light absorption in arteries and in veins, as the technique includes the measurement of the difference in light absorption between systole and diastole, which occurs because systole increases the volume of *arterial* blood. The absorption measurements in the cardiac-induced oscillations in the arterial blood volume also eliminate the effect of light attenuation by scattering [27] or by absorption into nail polish [28], because the latter attenuation effects are not oscillatory with the heart rate. The detailed theory of pulse oximetry that yields SpO_2_ from PPG parameters has been described in several publications [1,27,29]. In this study, only the aspects of the theory that are relevant to the explanation of the overestimation of SpO_2_ in hypoxemia will be presented. 

The analysis of the PPG signal includes a derivation of the ratio AC/DC for each pulse and each wavelength, where AC is the peak-to-peak amplitude of the oscillatory component of the pulse and DC is the mean light transmission during the pulse (Figure 2), and the ratio-of-ratios R, the ratio of AC/DC between the two wavelengths. Making use of Beer-Lambert Law, SaO_2_ is related to the value of R by [27,29,33,34]:(1)$${\mathrm{SaO}}_{2}=\frac{\varepsilon_{D1}-\left(\ell_{2}/\ell_{1}\;\right){R\varepsilon }_{D2}}{\left(\ell_{2}/\ell_{1}\right)R(\varepsilon_{O2}-\varepsilon_{D2})+(\varepsilon_{D1}-\varepsilon_{O1})}$$
where ε_D1_, ε_D2_, ε_O1_ and ε_O2_ are the values of the extinction coefficient for deoxy- and oxy-hemoglobin, respectively, for the two wavelengths λ_1_ and λ_2_. ℓ_1_ and ℓ_2_ are the optical pathlengths for λ_1_ and λ_2_, respectively. The differential pathlength factor (DPF), the ratio between the pathlength ℓ and the distance d between the light-emitter and the detector, depends on the reduced scattering constant µ′_s_ and the absorption constant µ_a_, according to Equation (2) [35,36]:ℓ(λ)= d × DPF(λ) = ½ d (3µ′_s_/µ_a_)^1/2^(2)
ℓ and DPF increase with µ′_s_ and decrease with µ_a_. Being dependent on µ′_s_ and µ_a_, ℓ and DPF depend on the composition and structures of the tissue and demonstrate intersubject variability [36]. The consequent intersubject variability in the pathlengths-ratio ℓ_2_/ℓ_1_ is a factor contributing significantly to the disparity between SpO_2_ and SaO_2_, as will be described in the following text. 

In order to obtain SaO_2_ from Equation (1) and the PPG-derived R, the pathlengths-ratio, ℓ_2_/ℓ_1_, for a given pair of wavelengths should be known, but ℓ_2_/ℓ_1_ varies between different subjects and cannot be measured individually, although a rough estimate of its *mean* value can be obtained from the literature [34]. The relationship between the required parameter SpO_2_ and the measured parameter R is therefore achieved in commercial pulse oximeters by calibration that includes simultaneous measurements of PPG-derived R and SaO_2_ in extracted arterial blood, measured by means of a CO-oximeter. The measurements are performed on healthy volunteers experiencing controlled desaturation achieved by reducing the fraction of oxygen in the inhaled air, yielding SaO_2_ values between 100% and 70%. From the acquired series of R/SaO_2_ pairs, an empirical calibration curve, SaO_2_ versus R, is created, and in pulse oximetry, for each measured R, SpO_2_ can be obtained from the curve [28,37,38,39]. The SaO_2_ versus R curve is specific to each pair of wavelengths; its characteristics also depend on the volunteers’ group. 

The calibration avoids the need to know the values of the extinction coefficients and the pathlengths-ratio, ℓ_2_/ℓ_1_, as required when Equation (1) is used. However, since the pathlength and the pathlengths-ratio vary among subjects, the SaO_2_ versus R relationship is based on the *average* pathlengths-ratio of the participants in the calibration process, and examinations of the SpO_2_ in individual patients will show a difference from SaO_2_, if their pathlengths-ratio differs from the average. The intersubject variability in ℓ_2_/ℓ_1_ is the origin of the standard deviation of the SpO_2_−SaO_2_ difference that generally occurs in clinical examinations. 

The intersubject variability in the pathlengths-ratio depends on the difference between the two wavelengths, λ_1_ and λ_2_, mainly because the scattering constant decreases monotonically with the wavelength in the red and near-infrared regions [40,41,42]. In particular, in the available commercial pulse oximeters, one of the two wavelengths is chosen in the red region (e.g., 660 nm) and the other one in the infrared region (e.g., 940 nm) so that the difference in the scattering constant and the pathlength between the two wavelengths are relatively large. The intersubject variability in the pathlengths-ratio is a consequence of the large difference between the pathlengths of the two wavelengths.

In studies by Nitzan et al. [32,43] the two wavelengths were chosen in the near-infrared region, with a relatively small difference between them (e.g., 761 and 820 nm). SpO_2_ was determined from R through Equation (1), assuming ℓ_2_/ℓ_1_ = 1, and using an empirical correction factor based on invasive SaO_2_ measured by CO-oximetry. In the more recent study [32], measurements were performed on preterm neonates and on children in a pediatric intensive care unit (PICU) that had a *normal* SaO_2_ range, i.e., above 90%. Despite the smaller difference between the two wavelengths and the corresponding smaller difference between the pathlengths of the two wavelengths relative to those of the commercial red-infrared pulse oximeter, no improvement in the accuracy was found, as was demonstrated by the standard deviation of the SpO_2_−SaO_2_ differences. In the current study, we compared SpO_2_ readings using a red-infrared commercial pulse oximeter and our two-infrared pulse oximeter for the invasive CO-oximetric measurement of SaO_2_ in cyanotic neonates in order to examine the effect of pathlengths-ratio on SpO_2_ measurement in *hypoxemic* infants. 

## 3. Materials and Methods

### 3.1. The Patients and the Clinical Examinations

A prospective study was conducted in the Neonatal Intensive Care Unit (NICU) of the Schneider Children’s Medical Center of Israel. Examinations were performed on 17 newborns (full term and preterm) with congenital heart diseases, admitted to the NICU before cardiac surgery or catheterization, whose oxygen saturation as measured by pulse oximetry was below 90%. The newborns had umbilical or peripheral post-ductal arterial catheter lines for clinical needs. The study was approved by the Institutional Review Board (Number 0886-18-RMC) and informed parental consent was obtained. 

The infants included in the study underwent the simultaneous measurement of SpO_2_ by two pulse oximeters: a commercial device, Nellcor™ Oximax Bedside SpO_2_ Patient Monitoring System (Medtronic Parkway, Minneapolis, MN, USA) that makes use of two wavelengths in the red and infrared regions (660 and 900 nm), and an experimental pulse oximeter, developed at the Lev Academic Center (Jerusalem, Israel), with two wavelengths in the infrared region (761 and 820 nm). The latter device will be described in more detail in the next section. The sensors of the devices were attached to different feet of the newborns. Invasive SaO_2_ in extracted arterial blood was measured using a blood gases analyzer (RAPIDPoint^®®^ 500e blood Gas System, Siemens Healthineers, Tarrytown, NY, USA). 

In this study, only the first triad of the SpO_2_ and SaO_2_ measurements on each infant was used, as our comparison of the performance of the two pulse oximetric techniques was based on the common statistical methods, which require independent data, while the results of two examinations on the same person are not independent when the sample includes different hypoxemic patients. This subject will be examined further in the Section 5.

### 3.2. The Two-Infrared Pulse Oximeter and Determination of the SpO_2_

The pulse oximeter with two wavelengths in the infrared region was designed and constructed at the Lev Academic Center—Jerusalem College of Technology (JCT). The JCT pulse oximeter consisted of an optical sensor, an electronic unit, and a tablet for the display of the PPG signals and their offline analysis. The sensor was a Nellcor pulse oximeter strip sensor for neonates (Nellcor compatible, Med Linket, China) in which the dual red-infrared light emitting diode (LED) was replaced by a dual LED (SLP, Israel), emitting light in two wavelengths in the near-infrared range (761 and 820 nm). The original photodetector of the Nellcor sensor was used for the measurement of the PPG pulses. The analysis of the PPG signals for the determination of the SpO_2_ was performed by MATLAB. The details of the control unit of the JCT pulse oximeter and the data analysis were described in our previous article [32]. 

The SpO_2_ was determined by using Equation (1), assuming ℓ_2_/ℓ_1_ = 1. Based on the Lambert-Beer equation, R was not taken as AC/DC ratio for the two wavelengths, but defined as the ratio of ln(I_D_/I_S_) for the two wavelengths, where I_D_ and I_S_ are the light intensity at the maximum (end-diastole) and minimum (end-systole) of the PPG pulse (Figure 2) (see [32]). The use of the ratio of ln(I_D_/I_S_) for the two wavelengths for R is considered more accurate than the use of AC/DC, though the difference in R calculation between the two techniques is generally small when AC/DC < 3% [32]. 

As was found in other studies (and as mentioned above), the SpO_2_ was overestimated in comparison to the SaO_2_ in hypoxemia. In order to reduce this disparity, correction/calibration of the readings of the two pulse oximeters was performed. The correction was based on the regression lines of the SpO_2_−SaO_2_ difference in the modified Bland-Altman plot (shown in the Section 4), which presents the relationship between SpO_2_−SaO_2_ and SaO_2_. The detailed process of the correction/calibration is described below in the Section 4 after the presentation of the modified Bland-Altman plot.

As described by Nitzan et al. [43], the PPG pulse maximum, I_D_, fluctuates at low and high frequencies, and the value of the PPG pulse maximum at the time of the apparent pulse peak (end-diastole) might differ from the value that would have been observed at the time of the pulse minimum. In deriving the amplitude of a given pulse, the value of I_D_, used for a given pulse, was taken as the value of the point of the line that connects the two maxima of the pulse at the time of the pulse minimum (see Figure 7 in Nitzan et al. [43], for a schematic illustration).

### 3.3. Statistical Methods

As explained above, only the first triad of the SpO_2_ and SaO_2_ measurements on each infant was used in order to eliminate statistical biases due to the performance on a single patient of multiple measurements, which are not independent when the sample includes several patients. The bias is greater when a different number of examinations are performed on each patient.

Data analysis of the SpO_2_ and SaO_2_ readings (after performing the calibration mentioned above) included the calculation of the mean, the standard deviation (SD), and the root mean square of the difference between the SaO_2_ and SpO_2_ readings (either those based on wavelengths in the red and infrared regions or those based on two infrared wavelengths) for the purpose of comparing the performance of the two pulse oximetric techniques. The correlation coefficient between the SaO_2_ and the SpO_2_ (obtained by both techniques) was also calculated for the comparison of the performance.

The comparison of the standard deviation of the SpO_2_−SaO_2_ differences for the two techniques was carried out by performing an F-test. The statistical analyses were performed using the IBM SPSS statistical package (SPSS 24, Chicago, IL, USA). 

Statistical significance was considered if *p* < 0.05.

## 4. Results

Examinations were performed on 17 children with cyanotic CHD. Table 1 presents the readings of the SaO_2_ and SpO_2_ obtained by emitted red and infrared light (SpO_2__R-IR) and by light with two wavelengths in the infrared region (SpO_2__2IR). The SpO_2_−SaO_2_ differences for both techniques are also presented. The average of SpO_2_−SaO_2_ differences was positive, as expected in hypoxia because of the overestimation of the SpO_2_. The standard deviation for the SpO_2__R-IR was 5.5%, and for SpO_2__2IR it was 4.5%. The correlation coefficient between the SaO_2_ and SpO_2_ established by the two techniques was 0.83 and 0.94, respectively. Note that the values of SpO_2__2IR were obtained by Equation (1), assuming ℓ_2_/ℓ_1_ = 1, before the correction/calibration that will be described below.

Figure 3 presents the relationship between SpO_2_ measurements using each of the two pulse oximeters and the corresponding SaO_2_, and the regression lines for the two functions. The value of R-squared (R^2^) for SpO_2__R-IR and for SpO_2__2IR was 0.68 and 0.88, respectively. R^2^ estimates the proportion of variance in the SpO_2_ that can be explained by SaO_2_ and is a measure for the goodness of the data fit to the linear regression model. Figure 4 presents modified a Bland-Altman plot of the SpO_2_−SaO_2_ difference for the two SpO_2_ techniques as a function of SaO_2_. The SpO_2_−SaO_2_ difference is small for SaO_2_ above 80%, but is about 12% for low values of SaO_2_, demonstrating an overestimation of SpO_2_ for patients with lower oxygen saturation. The value of R^2^ for SpO_2__R-IR and for SpO_2__2IR was 0.58 and 0.78, respectively.

Table 1 also shows values of SpO_2_ that were corrected to respond to the overestimation of SpO_2_ in low values of oxygen saturation. The correction is based on the regression lines in the modified Bland-Altman plot in Figure 4, which presents the relationship between SpO_2_−SaO_2_ and SaO_2_, and is performed through the coefficients a and b of the regression lines of for each of the pulse oximetric techniques:SpO_2_−SaO_2_ = a × SaO_2_ + b → SaO_2_ = (SpO_2_ − b)/(a + 1).(3)

Figure 5 presents the modified Bland-Altman plot, SpO_2_−SaO_2_ versus SaO_2_ for SpO_2__R-IR and SpO_2__2IR, after the calibration described by Equation (3). The average of the SpO_2_−SaO_2_ differences that was positive and relatively high for SpO_2__R-IR and SpO_2__2IR became zero after the correction, as expected. The standard deviation of the individual differences SpO_2_−SaO_2_ for SpO_2__R-IR increased to 6.5%, and for SpO_2__2IR it decreased to 3.6%. After the correction, the difference in the standard deviation between the two pulse oximeters was statistically significant. (*p* < 0.05 in F-test for standard deviations).

In summary, our results show that SpO_2_ that is based on two wavelengths in the infrared region better estimates SaO_2_ than SpO_2_ that is based on wavelengths in the red and infrared regions, by three parameters: the standard deviation of SpO_2_−SaO_2_ for SpO_2__2IR is lower than that for SpO_2__R-IR (3.6 versus 6.5%, respectively), the correlation of SpO_2__2IR with SaO_2_ is higher than that of SpO_2__R-IR with SaO_2_ (0.94 and 0.83, respectively), and the value of R-squared, which estimates the fitness of the data to the linear regression model, is higher for SpO_2__2IR than for SpO_2__R-IR (0.88 and 0.68, respectively). Another parameter, A_rms_, the root mean square (rms) of the difference between SpO_2_ and SaO_2_, was suggested by the ISO committee [44] and by Batchelder and Raley [45]. In general, A_rms_ is larger than the standard deviation, as it is affected both by random scatter and by the mean bias [44,45]; however, in our study, the mean bias between SpO_2_ after the calibration and SaO_2_ is zero for both pulse oximeters (Table 1), so that A_rms_ and SD are similar: 3.5 versus 6.3% for SpO_2__2IR and for SpO_2__R-IR, respectively.

The summary of these parameters is presented in Table 2.

## 5. Discussion and Conclusions 

In Section 2.1 in the Background section, we described studies on adults and neonates in hypoxemia, in which SpO_2_ overestimated actual arterial oxygen saturation in a significant percentage of the examinations performed. Those studies were performed with healthy volunteers, and they used commercial pulse oximeters based on the measurement of PPG signals generated by light with a pair of wavelengths in the red and infrared regions after calibration. The overestimation of SaO_2_ in SpO_2_ measurements in hypoxemia was also discovered in our study, where the two wavelengths were chosen in the infrared region, and where the SpO_2_ was derived from Equation (1), assuming a pathlengths-ratio ℓ_2_/ℓ_1_ = 1. The SpO_2_ that was based on two wavelengths in the infrared region had better correlation with SaO_2_ than SpO_2_ that was based on wavelengths in the red and infrared regions. After suitable calibration, utilizing the regression lines of the individual SpO_2_−SaO_2_ differences versus SaO_2_ in the modified Bland-Altman plot for the two pulse oximeters, the two-infrared pulse oximeter showed lower standard deviation than the red-infrared one.

Equation (1) shows that the relationship between the PPG-based parameter R and SaO_2_ depends on the pathlengths-ratio, ℓ_2_/ℓ_1,_ and the extinction coefficients. The extinction coefficients for HbO_2_ and deHb for a given pair of wavelengths are invariable among subjects, and only the pathlengths-ratio can change the relationship between R and SaO_2_. In the following, it will be shown that lower oxygen saturation is accompanied by a higher ratio of the absorption constants for red and infrared light, µ_a1_/µ_a2_, which results in an increase in the optical pathlengths-ratio, ℓ_2_/ℓ_1_. It will also be shown that an increase in ℓ_2_/ℓ_1_ may explain the disparity between SpO_2_ and SaO_2_ found in hypoxemia. These relationships are presented in titles in Figure 6, and detailed explanations are contained in the following text.

In order to understand the relationship between hypoxemia and the change of the ratio ℓ_2_/ℓ_1_, we should utilize Equation (2), which shows that the pathlength depends on the ratio (µ′_s_/µ_a_)^1/2^. When oxygen saturation changes, the light *absorption* of the hemoglobin molecules changes owing to the change in the relative concentrations of HbO_2_ and deHb molecules, but the *scattering* constant for the red blood cells is the same for HbO_2_ and deHb in the red and infrared regions [40]. Hence, the change in the pathlengths-ratio due to the change in SaO_2_ depends only on the absorption constants ratio, as ℓ_2_/ℓ_1_ is proportional to (µ_a1_/µ_a2_)^1/2^. In pulse oximetry the two wavelengths, λ_1_, λ_2_ (λ_1_ < λ_2_), are chosen in the two sides of the isosbestic point (see Figure 1) so that µ_a1_, the absorption constant for the wavelength that is shorter than the isosbestic point, *increases* in low oxygen saturation when the deHb concentration in the arterial and venous blood increases and HbO_2_ concentration decreases. Similarly, µ_a2_, the absorption constant for the wavelength that is larger than the isosbestic point, *decreases* in low oxygen saturation because the extinction coefficient for HbO_2_ is smaller than that for deHb (Figure 1). Hence, (µ_a1_/µ_a2_)^1/2^ increases, and consequently ℓ_2_/ℓ_1_ also increases in hypoxemia.

In our pulse oximetric technique, in which the two wavelengths in the infrared region are chosen, SpO_2_ is obtained by measuring the parameter R and substituting it in Equation (1), neglecting the factor ℓ_2_/ℓ_1_ by assuming that ℓ_2_/ℓ_1_ = 1. As SaO_2_ in Equation (1) is a *decreasing* function of ℓ_2_/ℓ_1_, and as ℓ_2_/ℓ_1_ in hypoxemia increases, the elimination of ℓ_2_/ℓ_1_ by our pulse oximetric technique results in an increase in SpO_2_ relative to SaO_2_ when SaO_2_ decreases, as can actually be seen in Figure 4. 

The Nellcor pulse oximeter that was used in our study showed a similar increase in the SpO_2_−SaO_2_ differences for decreasing SaO_2_ (Figure 4), and similar behavior of Nellcor SpO_2_ readings were found in the studies by Kim [4] and Harris [5]. However, the former explanation of the increase of the SpO_2_−SaO_2_ differences for lower values of SaO_2_ cannot be applied to the Nellcor pulse oximeter, owing to the calibration of the Nellcor pulse oximeter during its manufacture. As described in the background, the SaO_2_ versus R curve is achieved by simultaneous measurements of PPG-based R and SaO_2_ performed in healthy adult volunteers experiencing controlled desaturation; these measurements yield SaO_2_ values between 100% and 70%. The empirical calibration curve of SaO_2_ versus R is created from the acquired series of R-SaO_2_ pairs, and in pulse oximetry, for each measured R, SpO_2_ is obtained from the curve [28,37,38,39]. Since the calibration process also included low values of SaO_2_, it was expected that the pathlength ratio, ℓ_2_/ℓ_1_, would also increase in these cases; therefore, it should become evident why measurements taken using pulse oximetry in clinical examinations provided SpO_2_ values higher than the SaO_2_ values.

The overestimation of SaO_2_ by SpO_2_ in *clinical* examinations of patients with hypoxemia, despite the calibration that included SaO_2_ values down to 70%, can be explained by the different subjects examined in the calibration process and in the clinical examinations. The calibration is performed on healthy adult persons who experience controlled acute desaturation for only a relatively short time, whereas the clinical examinations are performed on sick patients, in whom certain relevant physiological functions might be compromised. The SpO_2_ measurement in sick patients with hypoxemia might have been affected by factors relating to their medical conditions; healthy subjects were probably not affected in this way. A possible explanation could be reduced perfusion to the peripheral organs, which might occur in some sick patients, and which results in lower oxygen saturation in *venous* blood [34,46] and consequently higher pathlength ratio and higher disparity between SpO_2_ and SaO_2_, as demonstrated in Figure 6. Further studies are required in order to better understand the relevant effects and substantiate this hypothesis.

The important outcome of this study is the greater correlation between SaO_2_ and SpO_2_ (0.94) when the latter is measured by two wavelengths in the infrared while SpO_2_ measured by two wavelengths in the red and infrared regions (0.83). The lower SD of SpO_2_−SaO_2_ that is obtained by the former, after proper correlation, relative to the latter (Table 2) seems to be attributable to the greater variability in the pathlengths-ratio in the red/infrared pulse oximeter. The greater difference in the absorption constant between photons with wavelengths in the red and infrared regions and the subsequent greater difference between ℓ_2_ and ℓ_1_ for photons with wavelengths in the red and infrared regions result in greater variability in the pathlength ratio, ℓ_2_/ℓ_1_. In addition, the pathlength is increased as the scattering constant µ′_s_ is reduced (Equation (2)), and the latter decreases monotonically with the wavelength in the red and infrared regions [40,41,42]. Hence, the greater difference between the red and infrared wavelengths relative to the difference in the two infrared wavelengths results in a greater difference in the scattering constant and consequently greater intersubject variability in the pathlengths-ratio, ℓ_2_/ℓ_1_. The greater intersubject variability in the pathlengths-ratio in the technique with red and infrared wavelengths seems to be the source of the greater standard deviation of the red and infrared pulse oximeter relative to the one using two wavelengths in the infrared region.

A limitation of the two-infrared technique relative to that using the red and infrared wavelengths is a smaller signal that might lead to a smaller signal-to-noise ratio (SNR). The signal in SpO_2_ measurement is R, the ratio of AC/DC between the two wavelengths, which is a measure of the difference between the amplitudes of the two PPG pulses in the two wavelengths. The parameter that mainly determines the R value is the difference in extinction coefficient between the oxygenated and deoxygenated hemoglobin in the wavelength range below the isosbestic point, which is greater for wavelengths in the red region (e.g., 665 nm) than in the infrared region (e.g., 760 nm). On the other hand, the main noise in pulse oximetry is caused by the light-scattering difference between the two PPG wavelengths; the scattering is also greater for wavelengths in the red region (e.g., 665 nm) than in the infrared region (e.g., 760 nm). The greater signal for a wavelength in the red region is counterbalanced by the lower noise in the infrared wavelength region.

The standard deviation of the SpO_2_−SaO_2_ differences after calibration using Equation (3) was 3.6% for the two-infrared pulse oximeter and 6.5% for the red-infrared pulse oximeter (5.5% before the current calibration) (Table 1). In other studies on cyanotic CHD children, the SD of the differences SpO_2_−SaO_2_ was 13.8% [19] and 6.6% [20], both measured using Masimo SET^®®^ LNCS Neo (Masimo, Irvine, CA, USA), and 7.8, 7.2 or 9.4% using Masimo Blue, Masimo LNCS, or Nellcor, respectively [4]. The variability in the SD is probably due to differences among the patient groups, and it also depends on the brand of the pulse oximeter used [4,5]. The differences between various brands of pulse oximeters seems to be related to the calibration process, which includes the selection of the healthy volunteers, who might differ in their pathlength ratio, ℓ_2_/ℓ_1_. The standard deviation of the individual SpO_2_−SaO_2_ differences in our two-infrared pulse oximeter after the correction (3.6%) is small relative to the studies cited above, but the comparison is not justifiable, because of two significant differences between the other three studies and ours. In our study, SpO_2_ readings (from both pulse oximeters) were corrected by regression lines that were created using the same data (in order to eliminate the bias), while in the three other studies, the calibration was carried out by the manufacturer on healthy volunteers, and the study was performed on *different* persons who were cyanotic CHD children. The other difference between the studies is that the three other studies were performed with multiple examinations on each patient (527, 515 and 258 paired SpO_2_ and SaO_2_ measurements on 25, 19 and 78 children, respectively), which significantly reduced the intersubject variability. In our study, a single examination was performed on each patient.

The advantage of the two-infrared over the red-infrared pulse oximeter is, however, demonstrated by its higher correlation coefficient with SaO_2_ and lower SD after the calibration, relative to the red-infrared pulse oximeter (Table 2), where the readings were obtained in the same study, in which a single examination was performed on each patient and in both cases the correction was performed by means of the regression line of the same examination data. The correction significantly reduced the standard deviation of the individual SpO_2_−SaO_2_ differences for the two-infrared pulse oximeter, while the SD for the red/infrared pulse oximeters slightly increased. 

In conclusion, pulse oximetry that makes use of two wavelengths in the infrared region appears to provide better estimates of SaO_2_ than red-infrared pulse oximetry, and it is suggested as a potential technique that might reduce the overestimation of SpO_2_ in hypoxemia. The common practice of calibration on *healthy* volunteers undergoing desaturation might lead to overestimation and lower accuracy of the SpO_2_ readings in clinical hypoxemia. The preliminary study that is presented in this article should be extended to a more comprehensive study, with a greater number of hypoxemic patients divided into correction and validation groups, and incorporating a single pair of examinations on each of them.

## Figures and Tables

**Figure 1 sensors-23-01434-f001:**
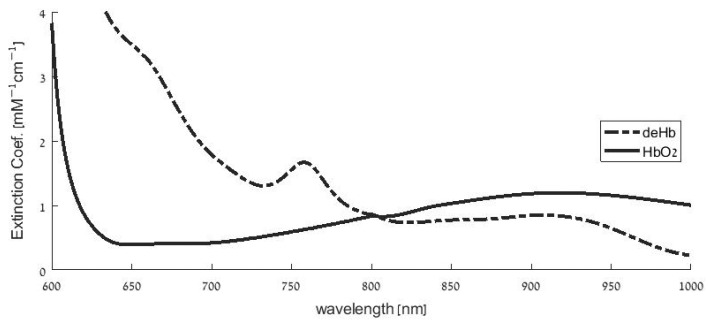
Extinction coefficient curves of deoxyhemoglobin (deHb) and oxyhemoglobin (HbO_2_) in the wavelength region of red and near-infrared light. The curves are based on data obtained by Zijlstra et al. (as presented in articles by Kim and Liu [30,31]) after cubic spline interpolation [32]. To the left of the isosbestic point (the intersection of the two curves at about 800 nm), the extinction coefficient for deHb is greater than that for HbO_2_, and vice versa to the right of it.

**Figure 2 sensors-23-01434-f002:**
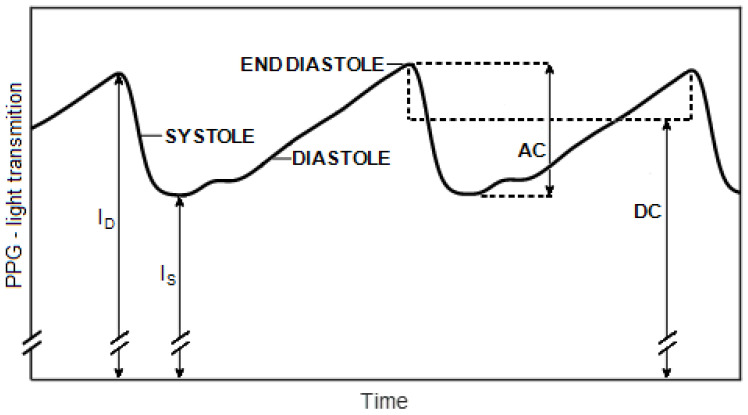
The PPG pulses—the transmitted light through the tissue as a function of time. The PPG signal decreases during systole, when blood enters the tissue, and increases during diastole. AC is the peak-to-peak amplitude—the difference between the maximum and minimum of the PPG pulse. DC is the mean light transmission throughout the PPG pulse. I_D_, I_S_—the transmitted light intensity at the maximum and the minimum of the PPG pulse, respectively.

**Figure 3 sensors-23-01434-f003:**
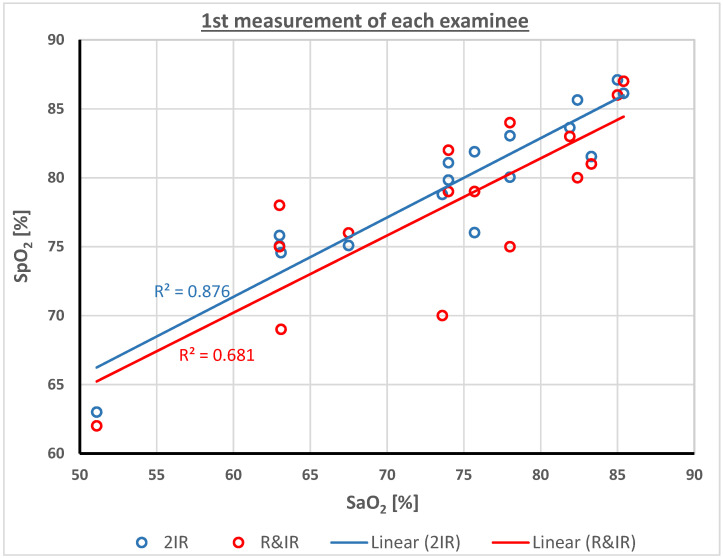
SpO_2_ readings as a function of SaO_2_ for SpO_2__R-IR (red circles) and SpO_2__2IR (blue circles). The lines are the respective regression lines.

**Figure 4 sensors-23-01434-f004:**
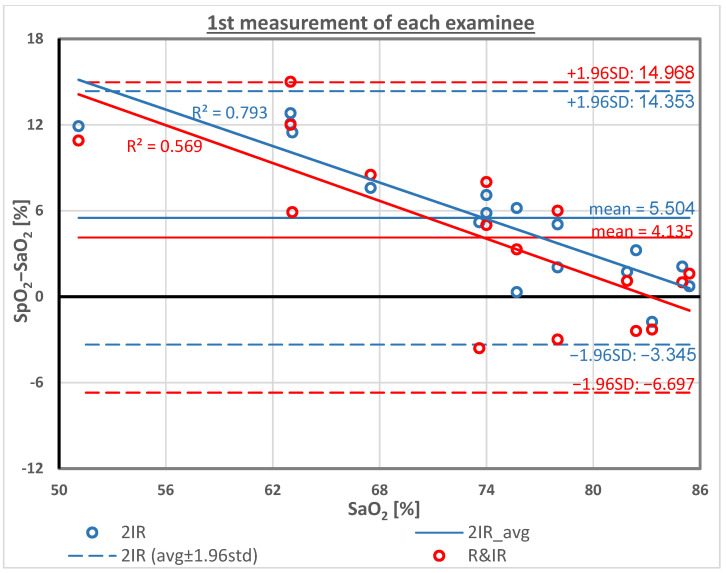
Modified Bland-Altman plot: SpO_2_−SaO_2_ as a function of SaO_2_ for the two SpO_2_ measurement techniques. SpO_2_−SaO_2_ is small for SaO_2_ above 80% but is about 12% for low values of SaO_2_.

**Figure 5 sensors-23-01434-f005:**
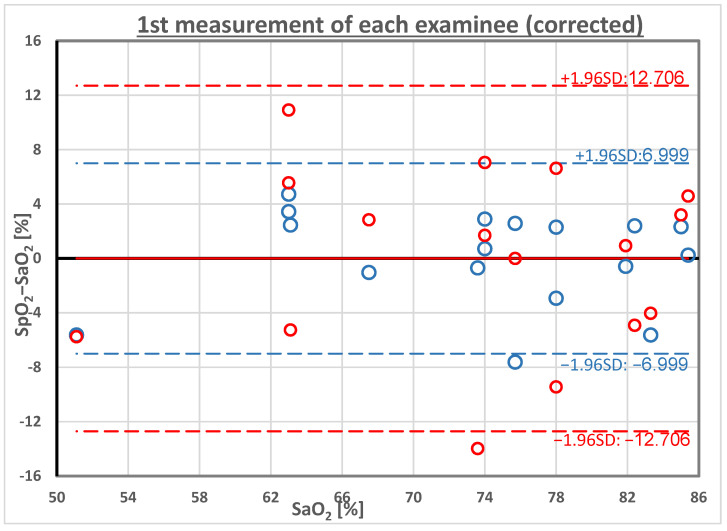
The modified Bland-Altman plot: SpO_2_−SaO_2_ as a function of SaO_2_ for the two SpO_2_ measurement techniques, after the calibration of Equation (3). The greater limits of agreement for the SpO_2__R-IR (red circles) relative to those for SpO_2__2IR (blue circles) reflect the difference in their standard deviations: 6.5% versus 3.6%, respectively.

**Figure 6 sensors-23-01434-f006:**
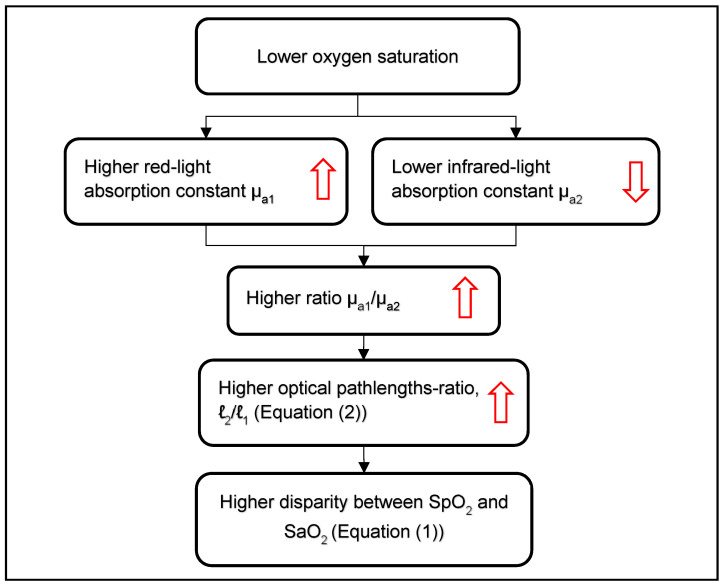
Flowchart describing the relationship between hypoxemia and the overestimation of SaO_2_ by SpO_2_. The steps are described in detail in the text.

**Table 1 sensors-23-01434-t001:** Readings of SaO_2_ and of SpO_2_, obtained by emitted red and infrared light (SpO_2__R-IR) and by light with two wavelengths in the infrared region (SpO_2__2IR). Corrected SpO_2__R-IR and SpO_2_-2IR are also presented. The average and the standard deviation (SD) of SpO_2_−SaO_2_ are also shown.

Pat. #	SaO_2_ [%]	SpO_2_ [%]		SpO_2_−SaO_2_ [%]		SpO_2_ Corrct. [%]		SpO_2_−SaO_2_ Corrct. [%]	
		R&IR	2IR	R&IR	2IR	Nellcor	2IR	Nellcor	2IR
1	83.3	81	81.5	−2.3	−1.8	79.3	77.7	−4.0	−5.6
2	63.1	69	74.6	5.9	11.5	57.8	65.6	−5.3	2.5
3	63.0	78	75.1	15.0	12.1	73.9	66.4	10.9	3.4
4	82.4	80	85.6	−2.4	3.2	77.5	84.8	−4.9	2.4
5	63.0	75	75.8	12.0	12.8	68.6	67.7	5.6	4.7
6	78.0	75	80.0	−3.0	2.0	68.6	75.1	−9.4	−2.9
7	67.5	76	75.1	8.5	7.6	70.3	66.5	2.8	−1.0
8	73.6	70	78.8	−3.6	5.2	59.6	72.9	−14.0	−0.7
9	85.0	86	87.1	1.0	2.1	88.2	87.3	3.2	2.3
10	51.1	62	63.0	10.9	11.9	45.3	45.5	−5.8	−5.6
11	74.0	82	81.1	8.0	7.1	81.1	76.9	7.1	2.9
12	75.7	79	76.0	3.3	0.3	75.7	68.1	0.0	−7.6
13	85.4	87	86.1	1.6	0.7	90.0	85.6	4.6	0.2
14	74.0	79	79.8	5	5.8	75.7	74.7	1.7	0.7
15	78.0	84	83.0	6	5.0	84.6	80.3	6.6	2.3
16	81.9	83	83.6	1.1	1.7	82.8	81.3	0.9	−0.6
17	75.7	79	81.9	3.3	6.2	75.7	78.3	0.0	2.6
Mean	73.8	77.9	79.3	4.1	5.5	73.8	73.8	0.0	0.0
SD	9.5	6.4	5.8	5.5	4.5	11.5	10.1	6.5	3.6

**Table 2 sensors-23-01434-t002:** Comparison of SpO_2__2IR or SpO_2__R-IR versus SaO_2_.

	SpO_2__R-IR (%)	SpO_2__2IR (%)
Standard deviation of SpO_2_−SaO_2_ after calibration *	6.5	3.6
A_rms_, the rms of SpO_2_−SaO_2_ after calibration	6.3	3.5
Correlation coefficient of SpO_2_ with SaO_2_ **	0.83	0.94

**p* < 0.05, ** *p* < 0.1.

## Data Availability

Data supporting reported results can be requested from Meir Nitzan, nitzan@g.jct.ac.il.

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
