# Peer review of "Overestimation of Oxygen Saturation Measured by Pulse Oximetry in Hypoxemia. Part 1: Effect of Optical Pathlengths-Ratio Increase"

_sensors, 2023, doi:10.3390/s23031434_

Round 1
Reviewer 1 Report
With the background of the current discussion about the accuracy of pulse oximeters in blacks, this is an interesting approach that should be pursued. Whether the optical path lengths alone are responsible for the difference must be further clarified. Especially for tissue with a high absorption, the perfusion index between the red and infrared LED has to be considered. The absorption of the red wavelength is significantly greater and there is a different ratio between R/IR. If the perfusion is now even lower, this has an even more dramatic effect on red and the resolution limit of the AD converter is reached. I recommend to additionally consider the perfusion index for both wavelengths in the measurements, this would be an interesting information for the reader and allows to draw further conclusions.
Reviewer 2 Report
The authors address the overestimation of oxygen saturation in hypoxemia by the example of cyanotic newborns.
In the present state, the manuscript has several shortcomings that need to be addressed and resubmitted for review prior to publication. Besides, the manuscript significantly suffers from lack of rigorous separation between the different sections (i.e., parts of the methods can be found in the results or even discussion section) and needs complete rework.
- The abstract must be completely revised as it is currently too vague. Please include specific details about the methods (e.g., number of subjects, type of measurement device, type of reference device, LED wavelengths, etc.) and the results (e.g., the resulting errors/performances in numbers), etc.
- SpO2 performance shall be measured by means of the ARMS as required by the ISO 80601-2-61 standard (Medical Electrical Equipment - Part 2-61: Particular Requirements for Basic Safety and Essential Performance of Pulse Oximeter Equipment. In my opinion there is not much interest in measuring R2 (line 252) for SpO2 performance but rather the abovementioned ARMS.
- Again, the outcome of the study should not be quantified by a correlation but by ARMS (lines 364-366, etc.).
- It is unclear to me why the authors only show the 1st measurement of each subject in Figures 3, 4, and 5. Please add all measurements and compute the figures of merit / performances on all measurements.
- Line 248/Table 1: the method for obtaining “corrected” SpO2 readings is not described in the methods section which must be done. When reading the beginning of the results section it is totally unclear to the reader what these “corrected” readings are. Please add a small section describing how the performances of the two devices was performed. There seems to be some explanation on line 268 (until line 272 including equation (3)) which does not go into the results but into the methods section. Please RIGOROUSLY revise your ENTIRE manuscript regarding these aspects, e.g., the results section shall not contain any methods or discussion, etc.
- On the same topic, line 400/401 states that only one examination was performed per patient in the current study. This is at the end of the discussion(!) section and until then I have not found this information in the text (I might have missed it). This clearly goes into the methods section.
- Only in the discussion section the authors clarify that their device was calibrated on the same data (which obviously leads to good performances) whereas the reference devices were calibrated during manufacturing. Again, this needs to be clearly stated in the methods section but can be addressed again as a limitation in the discussion.
- The previous point and the 2nd-last paragraph in the discussion (lines 402-409) reveal the flaw of the present study. In my opinion the authors cannot compare the performance of a clinical device (calibrated during manufacturing) vs. their own device calibrated on the same data as they estimate the performance. In addition, I do not see how the calibration coefficient is a reliable measure to prove that the two-infrared approach outperforms the read-infrared approach.
- The authors state that SpO2 measurements in sick patients are affected by other factors than in healthy subjects (lines 360-363) without providing examples, hypotheses of possible factors. Please at least provide a few possible factors, else this paragraph is too vague.
- The flowchart in Figure 6 does not seem to be the most appropriate means to describe the intended relationships between hypoxemia and SpO2 overestimation. How about using the equation(s) and illustrating (e.g., with arrows) the increase/decrease of different factors? (This is only a suggestion, no obligation).
- The authors should compare their suggested approach to other approaches e.g., by Badiola et al. (I. Badiola et al. Physiol. Meas., vol. 43, 2022, doi: 10.1088/1361-6579/ac890c) using the standard red-infrared wavelengths but different calibration/algorithms.
- The authors should clearly state the potential limitations related to using their proposed approach (two-IR instead of the standard red-IR) in terms of signal quality (e.g., lower SNR), availability/efficacy of LEDs, etc.
Minor comments:
- Line 16: « in hypoxemia » either remove (repetition) or move it to the end of the sentence.
- Line 204: please provide the IRB number/identificatory of the study.
- Line 208/209: please provide the exact red/infrared wavelengths of the Nellcor device.
- Line 209/210: please provide the exact wavelengths of your device. Probably good to mention that this device will be described in more detail in the next section.
- Line 218/219: please provide manufacturer, model/type of LEDs and photodetector.
- Line 294: which “background” are you referring to? Please rephrase/clarify.
- Line 301: please add the name/description (“path-lengths-ratio”) for l_2/l_1.
- Line 168: from my understanding the ISO standard / FDA guidance do not force you to do the calibration on healthy volunteers. If this is the case change “healthy volunteers “ to “subjects”.
- Line 210: remove “,” before “(Jerusalem[…]”
- Line 257: could you please quantify/test the significance of this “significant overestimation” by statistical methods/tests.
